# The Dispensing Error Rate in an App-Based, Semaglutide-Supported Weight-Loss Service: A Retrospective Cohort Study

**DOI:** 10.3390/pharmacy12050135

**Published:** 2024-09-03

**Authors:** Louis Talay, Matt Vickers

**Affiliations:** 1Faculty of Arts and Social Sciences, University of Sydney, Sydney, NSW 2050, Australia; 2Eucalyptus, Sydney, NSW 2000, Australia; matt@eucalytpus.vc

**Keywords:** dispensing errors, GLP-1 RA medications, digital pharmacy, obesity, chronic care, multidisciplinary care, third-party pharmacy, telehealth

## Abstract

Digital weight-loss services (DWLSs) combining pharmacotherapy and health coaching have the potential to make a major contribution to the global struggle against obesity. However, the degree to which DWLSs compromise patient safety through the dispensation of Glucagon-like peptide-1 receptor agonist (GLP-1 RA) medications is unknown. This study retrospectively analysed the rate at which patients reported GLP-1 RA dispensing errors from patient-selected and partner pharmacies of Australia’s largest DWLS provider over a six-month period. The analysis found that 99 (0.35%) of the 28,165 dispensed semaglutide orders contained an error. Incorrect dose (58.6%) and unreasonable medication expiry window (21.2%) were the two most common error types. Most errors (84.9%) were deemed to have been of medium urgency, with 11.1% being considered high-urgency errors. Incorrect doses (45.5%) and supplies of the wrong medication (36.3%) comprised most errors reported in high-urgency cases. Female patients reported more dispensing errors than male patients (0.41% vs. 0.12%, *p* < 0.001). Similarly, reported dispensing error rates were highest among patients aged 18 to 29 years (0.6%) and 30 to 39 years (0.5%). This research provides preliminary evidence that GLP-1 RA dispensing errors within comprehensive Australian DWLSs are relatively low.

## 1. Introduction

The incidence of obesity has become arguably the most serious global public health issue of the modern era. Overweight and obesity rates in Australia have been rising steadily since 1995 and the latest census data revealed that nearly two-thirds (65.8%) of the adult population live with excess weight [1]. To combat this problem, international health organisations stress the importance of ongoing, comprehensive care that is coordinated across a multidisciplinary team (MDT) [2,3]. However, managing mid- to long-term regular consultations in real-world, face-to-face (F2F) settings has historically been challenging for anyone with significant work or family commitments [4,5]. In countries with large regional populations such as the USA and Australia, many people living with obesity face an additional geographic barrier to quality care [6]. 

To overcome these access barriers, an increasing number of people are using digital weight-loss services (DWLSs) [7,8,9]. Several of these services are comprehensive, coordinating lifestyle and pharmacological therapy through MDTs, such as Ro in the USA and Eucalyptus in Australia, Germany, Japan, and the UK [7,10]. However, most countries with DWLSs have a broad care model spectrum, with those at the minimalist end offering little more than access to prescriptions for weight-loss medications, including Glucagon-like peptide-1 receptor agonists (GLP-1 RAs) [11]. While both Ro and Eucalyptus distinguish themselves from such care models by utilising GLP-1 RAs as supplements to MDT-guided lifestyle therapy [10,12], there is evidence that some patients simply use the services to facilitate access to these medications [5]. Moreover, both service providers arguably centre more marketing content around the medications rather than care continuity. For example, the Eucalyptus homepage for men’s weight loss presents the following statement in bold text: “What medical weight loss offers is a fair go at better health” [13]. The Ro weight loss homepage is arguably a little more subtle, stating, “When it comes to weight loss, biology is your nemesis. Not willpower” [14], whereby the reader logically infers from surrounding content (and perhaps recent media exposure) that GLP-1 RAs address the genetic component of weight management. In both cases, the medications are presented within a social equity frame. 

Consensus around obesity’s genetic component and the role of GLP-1 RAs in modifying neural pathways involved in appetite is relatively well established [15,16]. Additionally, clinical trials have consistently found GLP-1 RAs to be safe and highly effective in weight-loss cohorts [17,18]. From an effectiveness perspective, a recent study showed that a GLP-1-RA-supported DWLS generated a mean 5-month weight loss (9.52%) that was over three times higher than that from a standalone lifestyle intervention (3.1%) [19]. Despite these discoveries and the knowledge that standalone lifestyle interventions have a low success rate [20,21], medical institutions continue to raise questions about the way DWLSs provide their medications. Namely, they appear concerned that certain DWLSs have inadequate safety protocols and allow people unsuitable for GLP-1 RAs to obtain such medications [22,23]. This an appropriate concern given that the effects of GLP-1 RA weight-loss therapy on unsuitable patients (e.g., patients who are not overweight) are unknown. Moreover, although GLP-1-RA-supported DWLSs have demonstrated good weight-loss and adherence outcomes across various cultures [10,11,19,24,25], they are yet to present any meaningful safety-related data. Stakeholders who question DWLSs’ safety are ultimately concerned about errors that stem from one of two phases in the commencement of a patient’s care journey: the prescribing phase or the dispensing phase. 

Scholarly interest in medication errors appears to have risen over the past decade, some of which may be attributable to the World Health Organization’s “Medication Without Harm” campaign in 2017 [26]. However, in Australia, most of this research focuses on prescription-phase markers, such as preventable medication-related hospitalisations [27,28], adverse event figures [29], and inappropriate prescribing incidence [30,31]. To our knowledge, only one Australian peer-reviewed study on dispensing error rates has been published in the past decade—a 2016 study that detected an error rate of 11.5% in 3959 dose-dispensing aids used in 45 care facilities for aged patients [32]. Notably, a 2019 national Medicine Safety report from the University of South Australia and Pharmaceutical Society of Australia was unable to contribute dispensing error data despite acknowledging its critical role in reducing medication-related harm [33]. 

The best available benchmark for general dispensing safety appears to come from a global meta-analysis published in 2024. This meta-analysis pooled data from 62 studies across community, hospital, and other pharmacy settings and concluded that the worldwide prevalence of dispensing errors was 1.6% [34]. However, the study reported a dispensing error rate ranging from 0 to 33%, which is consistent with earlier meta-analyses and likely reflects considerable variance in pharmacy setting, technology use, and denominator type [35]. Of these three factors, denominator type had the greatest impact on the wide dispensing error range across the 62 studies. While some studies (n = 5, 8.1%) used the total number of treated patients as the error rate denominator, others used the total number of doses (n = 7, 11.3%) or dispensed items/orders (n = 45, 72.6%). In addition to the uncertainty around general dispensing error rates in Australian care settings, little is known about the degree to which digital modalities impact dispensing safety, both in Australia and across the rest of the world. Studies have demonstrated that electronic prescribing typically reduces medication error rates (thus, combined prescription, dispensing, and administration errors) in hospitals [36,37], but none appear to have analysed the impact on dispensation in isolation, let alone in non-hospital settings. Only one study identified in the meta-analysis by Um et al. assessed the safety of a remote service, reporting a slightly higher dispensing error rate (1.3%) in a group of telepharmacy sites versus a community comparator (0.8%) [38].

The safety of GLP-1 RA dispensation within DWLSs is becoming increasingly important, with large numbers of people accessing these medications through a broad spectrum of online providers. However. there is currently insufficient research in the literature to benchmark dispensing error rates in digital chronic care, let alone GLP-1 RA dispensing errors in DWLSs. Therefore, this study aims to analyse reported GLP-1 RA dispensing error rates for Australia’s largest comprehensive DWLS provider, Eucalyptus. 

## 2. Materials and Methods

### 2.1. Study Design

This investigation adopted a retrospective cohort study design to analyse a dataset of patients enrolled in the Juniper (women) or Pilot (men) Australia weight-loss programmes (both owned by Eucalyptus) between 1 June 2023 and 1 December 2023. This method was selected in accordance with the NHS Health Research Authority’s “Defining Research table” [39], having satisfied the following criteria: “designed and conducted solely to define or judge current care or service”; “measures current service without reference to a standard”; “involves analysis of existing data”; and “patient/service users have chosen intervention independently of the service evaluation”. Bellberry Limited approved the ethics of this study on 22 November 2023. 

### 2.2. Programme Overview

The Juniper and Pilot DWLSs have received accreditation through the Australian Council on Healthcare Standards and the UK Digital Technology Assessment Criteria [40,41]. Both deliver combined GLP-1 RA therapy and asynchronous health coaching via a mobile app and digital platform. All patients are allocated an MDT consisting of a physician, a university-qualified health coach, a pharmacist, and a medical support officer to guide them through a personalised weight-loss programme. Physicians determine patient eligibility for the Eucalyptus DWLS from pre-consultation questionnaires, which can contain over 100 questions, including requests for test results, clinician reports, and photos. 

Eligible patients are then provided with a GLP-1 RA prescription. Upon receiving payment for the weight-loss programme, the prescription is shared electronically with one of a range of partner community pharmacies for dispensation, unless the patient wishes to have the prescription dispensed at their own chosen pharmacy. Prescriptions are allocated to partner pharmacies based on the patient’s geographic location and stock availability, by way of a software-as-a-service (SaaS) platform dedicated to outsourced dispensation. 

All Juniper and Pilot patients are clearly instructed to inform their medical support officer via email or the programme’s in-app chat feature if the pharmacy makes any form of error (e.g., if they have distributed the wrong medication or labelled the order incorrectly). Reported errors are forwarded by medical support officers to the patient’s MDT, who consults the patient on the appropriate course of action. All error-related communications between patient, medical support staff, and clinicians are also forwarded to the Eucalyptus clinical auditing team, who store pharmacy error data in Jira software (version 9.14.x), including characterisation of the error type and urgency. Throughout the study period, semaglutide was the only GLP-1 RA medication prescribed to Eucalyptus Australia DWLS patients. All patients received 4 semaglutide syringes per order. The exact same dispensing and error reporting processes were used in the Juniper and Pilot DWLSs. A flow chart of the semaglutide dispensing process can be viewed in Figure 1.

### 2.3. Participants

This study’s cohort comprised all Juniper and Pilot weight-loss patients who received at least one semaglutide order between 1 June and 1 December 2023. Patient eligibility for either programme was determined by an Australian physician, who followed the semaglutide product information guidelines for weight-loss therapy [42]. The inclusion criteria were an initial body mass index (BMI) of 30 kg/m^2^ or greater (indicating obesity) or a BMI of 27 kg/m^2^ or greater (indicating overweight) in the presence of at least 1 weight-related comorbidity such as hypertension or sleep apnoea. Key exclusion criteria included the following contraindications: acute pancreatitis; a personal or family history of medullary thyroid carcinoma or patients with multiple endocrine neoplasia type 2; a previous acute kidney injury; acute gallbladder disease; hypoglycaemia; a severe mental health condition; known hypersensitivity to semaglutide or any of the product components; and patients with type 1 or type 2 diabetes. Eucalyptus physicians use their discretion in determining whether the semaglutide can be used concomitantly with other oral medications, which may interact with its gastric-emptying effect. Eucalyptus patients consented to the service’s privacy policy at the time of their subscription, which includes permission to use their de-identified data for research.

### 2.4. Procedures

Study data were retrieved from the Eucalyptus clinical auditing team’s issue-tracking repository on Jira software. The Eucalyptus clinical auditing team filtered all pharmacy errors reported by Juniper and Pilot weight-loss patients between 1 June and 1 December 2023 on Jira software and extracted them onto a csv spreadsheet for the study authors’ analysis. The total number of Eucalyptus semaglutide orders from the same 6-month period was used as the denominator. 

### 2.5. Endpoints

The primary endpoint was the dispensing error rate, which was calculated by dividing the total number of dispensing errors by the total number of semaglutide orders. This denominator was selected on the basis of its prevalence in the available literature on dispensing errors [19]. Error type (Table 1) and error urgency rating (Table 2) frequencies represented the study’s secondary endpoints. 

### 2.6. Statistical Analyses

Descriptive statistics are presented as means with standard deviations and the number of occurrences (percentages). Categorical variables (e.g., female-reported and male-reported dispensing error rates) were compared using chi-square tests. Analyses were conducted using RStudio, version 2023.06.1+524 (RStudio: Integrated Development Environment for R, Boston, MA, USA). 

## 3. Results

Between 1 June and 1 December 2023, the Eucalyptus DWLS dispensed 28,165 semaglutide orders to Australian patients. Baseline characteristics of these patients are reported in Table 3.

A dispensing error was reported by patients in 99 of the 28,165 orders over the study period, representing a total error rate of 0.35%. Female patients reported a slightly higher rate of dispensing errors (92 errors reported per 22,417 semaglutide orders (0.41%)) compared to male patients (7 errors reported per 5748 orders (0.12%)), χ^2^(1, N = 28,165) = 10.88, *p* < 0.001 (Table 4). Error rates were highest among patients aged 18–29 (0.6%) and 30–39 years old (0.5%), χ^2^(4, N = 28,165) = 10.5, *p* = 0.032. However, no significant associations were found between error rates and BMI groups (χ^2^(3, N = 28,165) = 5.2, *p* = 0.16) or ethnicity (χ^2^(1, N = 28,165) = 2.6, *p* = 0.08). Additionally, error rates did not differ between orders that were dispensed at patient-selected pharmacies and those that were dispensed at partner pharmacies allocated through an SaaS platform, χ^2^(1, N = 28,165) = 0.05, *p* = 0.83).

Incorrect dose (58.6%) and medication with an unreasonable expiry window (21.2%) were the two most common error types (Table 5). The vast majority of errors (84.9%) were deemed to have been of medium urgency, with 11.1% coded by the Eucalyptus auditing team as high-urgency errors. Dispensation of an incorrect dose (45.5%) or the wrong medication (36.3%) were the most commonly reported errors among high-urgency cases.

## 4. Discussion 

### 4.1. The Significance of Eucalyptus Australia’s GLP-1 RA Dispensing Error Rates 

This is the first study to report dispensing error rates in a GLP-1-RA-supported DWLS. Such services are becoming increasingly important throughout the world, as they arguably represent the most effective means of accessing continuous obesity care. Our analysis detected a low GLP-1 RA dispensing error rate for the Eucalyptus Australia DWLS. Most errors pertained to an incorrect dose or an unreasonable medication expiry window and were of medium urgency. A higher rate of errors was detected among females and patients aged 18–39 years old. 

The 0.35% dispensing error rate observed in this analysis appears to be low relative to other findings in the available literature [32,33]. However, as was discussed in the introduction, current data do not facilitate a clear comparison with dispensing error frequency in DWLSs. The only available Australian study on dispensing error rates was conducted in care services for aged patients [30]—a setting renowned for its high medication error rate. And although the recent meta-analysis by Um et al. gives a good indication of global dispensing error standards, only one study was found to assess remote services [36], which, in 2005, when the study was conducted, would have looked very different from a modern digital care service. It is reasonable to conceive of health stakeholders holding all care modalities to the same dispensing safety standard when data become more readily available. Until such time, it is difficult to evaluate the dispensing safety level of the Eucalyptus DWLS observed in this study. 

This study’s secondary endpoints shed light on some important considerations for DWLSs. The discovery that incorrect dose errors were the most reported error type and accounted for the largest proportion of high-urgency cases indicates that DWLSs should prioritise this problem in their efforts to minimise GLP-1 RA dispensing errors. This finding appears to be somewhat consistent with the meta-analysis by Um et al., which stated that “wrong medication strength” was among the most common error types reported in the literature on dispensing errors [34]. Integrating Eucalyptus dispensing data from partner pharmacies into the Eucalyptus clinical auditing system might also contribute to a safer service. At present, the service relies exclusively on patient reports to detect dispensing errors. In cases of incorrect dose errors, for example, patients may decide against reporting the error due to their perception of a desirable outcome (e.g., the perception that a higher dose = faster weight loss). If Eucalyptus auditors were notified of dispensing errors the moment they occurred, the error detection and response rates would likely increase. Although pharmacies may ultimately reject this level of integration on the basis of its legal and/or logistical challenges, DWLSs and comparable digital services should investigate its potential.

The statistically lower dispensing error rate observed among male patients is consistent with the findings of other research on self-reported medication errors [43,44]. However, as was the case in these earlier studies, we cannot draw any meaningful conclusions about the observed gender disparity, as errors were reported by patients rather than independent reviewers and no data were collected on the manner in which patients assessed their medication orders. Other research has indicated that, relative to women, men have a lower inclination for disclosing negative information [45] or engaging with healthcare in general [46]. The negative association observed between dispensing errors and age could possibly be explained by a higher degree of familiarity with app-based healthcare among younger patients [47], and thus a higher likelihood of such patients reporting errors. However, further research is needed to determine whether any intrinsic factors predispose younger adults to a higher rate of GLP-1 RA dispensing errors in DWLSs.

### 4.2. Public Health Implications

The above-discussed findings could have various public health implications. First and foremost, the observed dispensing error rates may be interpreted as preliminary evidence of the potential of GLP-1-RA-supported DWLSs to provide safe care. This will hopefully stimulate further research into other DWLS safety markers and efforts to develop national clinical governance standards for digital chronic care providers. Secondly, this study’s secondary endpoints may encourage the development of tighter monitoring mechanisms for digital care models that outsource medication dispensation. It was argued above that this responsibility should fall on DWLS providers, but it is feasible that this study and subsequent DWLS safety research will contribute to collaborative initiatives that promote transparency, such as real-time monitoring of GLP-1 RA prescription and dispensation. Finally, this study adds another layer of nuance to the emerging literature on GLP-1-RA-supported DWLSs [5,10,11,19,24,25], which could feasibly lead to greater stakeholder confidence, government subsidisation, and, in turn, better public health outcomes. It is clear that traditional weight-loss services are incapable of addressing the global obesity problem [17,18,19]. GLP-1-RA-supported DWLSs have shown promise in making a significant contribution to a comprehensive solution [5,8,17,22,23]. However, public health systems need clear evidence of the safety of such services before they fully embrace them. This study represents the preliminary step towards a body of evidence that will allow an informed public safety assessment. 

### 4.3. Strengths and Limitations

This study’s strengths included its sample size, its broad inclusion criteria, its non-interference with patient experience, and its novelty. In terms of its limitations, firstly, all data were self-reported, and thus subject to numerous patient biases and constraints. Secondly, a detailed investigation into the lower dispensing error rate among males was not possible and should be explored in future research. And finally, the error categories could have benefitted from additional nuance. For example, presenting error range sub-categories for incorrect dose (mg) and unreasonable medication expiry window errors (days) would have provided a clearer view of the risks in DWLS dispensation. Data on the exact degrees of such errors are currently not stored on the Eucalyptus issue-tracking repository. 

### 4.4. Future Research

Future research should aim to measure dispensing error rates in other GLP-1-RA-supported DWLSs in order to enable the eventual creation of an industry benchmark. Researchers should also start to analyse GLP-1 RA prescribing error rates in both DWLSs and F2F weight-loss programmes. Case studies on clinical responses to medication errors in GLP-1-RA-supported DWLSs would add another important layer to this emerging field of literature.

## 5. Conclusions 

The findings from this study lay a vital foundation for ongoing research on dispensing safety in DWLSs and digital care in general. Despite the increasing uptake of GLP-1-RA-supported DWLSs in recent years, the literature on the safety of such services remains scarce. GLP-1 RA dispensing errors represent one of the key dangers of such services, given both the relative potency of these medications and the outsourcing of their dispensation. Our findings suggest that comprehensive DWLSs have the potential to safely outsource GLP-1 RA dispensation to partner pharmacies via dedicated SaaS platforms. However, research to benchmark dispensing error rates and explore prescribing errors are needed to further support the safety of this care delivery model. 

## Figures and Tables

**Figure 1 pharmacy-12-00135-f001:**
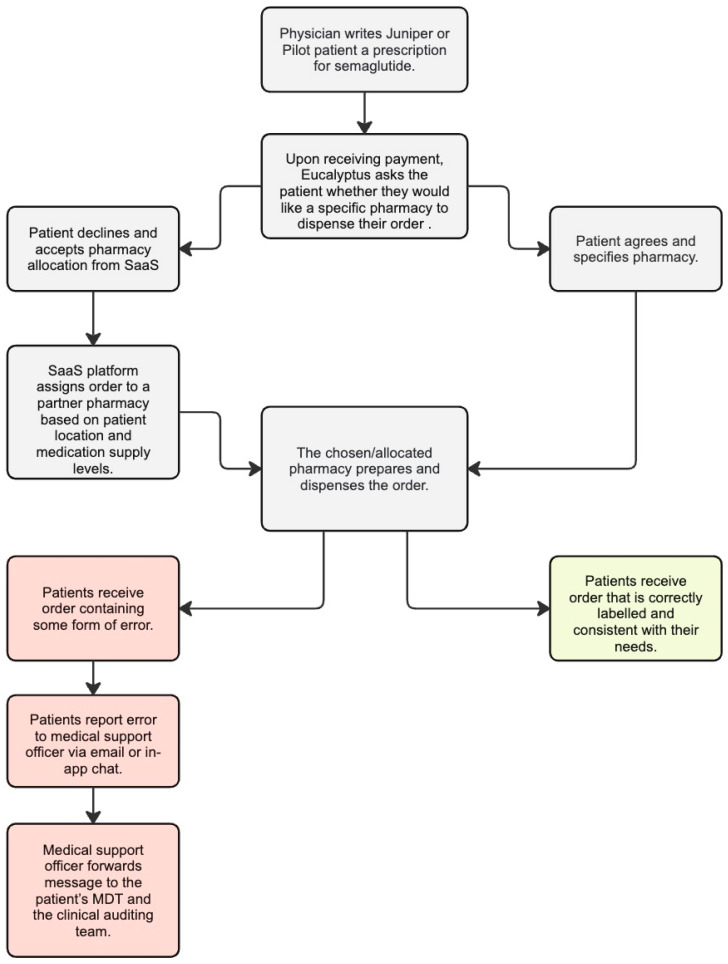
Flow chart of Juniper and Pilot semaglutide dispensing process, created with Miro software (version 0.7.37).

**Table 1 pharmacy-12-00135-t001:** Description of dispensing error types.

Error Type	Description
Wrong medication	The patient received a medication other than semaglutide or another patient’s order of semaglutide.
Labelling error	The patient received an order with an incorrect title, a spelling mistake, or any other error on the label.
Unreasonable medication expiry window	The patient received an order containing a syringe(s) that could not be administered before the stated expiry date when adhering to the prescribed dosing schedule.
Incorrect dose	The patient received an order with a dose of semaglutide that was inconsistent with their prescription.

**Table 2 pharmacy-12-00135-t002:** Description of dispensing error urgency ratings.

Urgency Rating	Description
High	An error that could result in major harm (temporary or permanent), including death, which is not reasonably expected as an outcome of healthcare.
Medium	An error likely to result in minor harm, which is not reasonably expected as an outcome of healthcare.
Low	An error likely to result in no harm or a “near miss” for the patient.

**Table 3 pharmacy-12-00135-t003:** Baseline characteristics of patients who received semaglutide during the study period.

Demographic Information	Mean (SD)
Age	42.67 (±8.74) years
**Gender**	**Number (%)**
Female	4912 (77.92)
Male	1392 (22.08)
**Ethnicity**	**Number (%)**
Caucasian	5178 (82.14)
Middle Eastern	442 (7.01)
Asian, including the subcontinent	347 (5.50)
Black African or African Caribbean	151 (2.40)
Latino/Hispanic	118 (1.87)
Rather not say	60 (0.95)
**Clinical information**	**Mean (SD)**
BMI	33.47 (±5.82) kg/m^2^
Weight	97.89 (±17.86) kg

**Table 4 pharmacy-12-00135-t004:** Chi-square analyses of dispensing errors according to age and BMI categories.

Category	Number of Errors (n)	Total Dispensed Items (n)	Error Rate (%)	*X* ^2^	*p*-Value
**Gender**				10.9	<0.001 ***
Female	92	22,417	0.4		
Male	7	5748	0.1		
**Age group (years)**				10.5	0.03 *
18–29	13	2322	0.6		
30–39	38	7913	0.5		
40–49	24	9697	0.2		
50–59	18	6396	0.3		
60+	6	1837	0.3		
**Body Mass Index group (kg/m^2^)**				5.2	0.16
27–29.99	23	6968	0.3		
30–34.99	37	12,914	0.3		
35–39.99	22	4434	0.5		
40+	17	3849	0.4		
**Ethnicity group**				2.6	0.08
Caucasian	81	21,063	0.4		
Non-Caucasian	18	7102	0.3		
**Dispensing** **pharmacy**				0.1	0.83
Patient-selected pharmacy	16	4335	0.37		
Partner pharmacy	83	23,830	0.35		

Notes: * *p* < 0.05, *** *p* < 0.001.

**Table 5 pharmacy-12-00135-t005:** Dispensing errors by type and urgency rating.

Error Type—No. (% of Total Errors)	
Incorrect dose	58 (58.6%)
Unreasonable medication expiry window	21 (21.2%)
Labelling error	14 (14.1%)
Wrong medication	6 (6.1%)
**Urgency rating—no. (% of total errors)**	
High	11 (11.1%)
Medium	84 (84.9%)
Low	4 (4%)
**Error type in high-urgency cases—no. (% of total high-urgency errors)**	
Incorrect dose	5 (45.5%)
Wrong medication	4 (36.3%)
Unreasonable medication expiry window	2 (18.1%)

## Data Availability

The data presented in this study are available from the corresponding author on reasonable request.

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
