# Peer review of "The Dispensing Error Rate in an App-Based, Semaglutide-Supported Weight-Loss Service: A Retrospective Cohort Study"

_pharmacy, 2024, doi:10.3390/pharmacy12050135_

Round 1

Reviewer 1 Report

Comments and Suggestions for Authors

 - In both the study title and the study design, the term "case study" should be replaced with "cohort study."

 - It would be interesting for readers to provide data on obesity prevalence in Australia;

 - Line 36 - I recommend including more information regarding services provided by Ro and Eucalyptus to present their main mission in а summary.

 - Line 59 and Line 70 - The Um et al. (2024)/The Um et al. (2023) - replace the year in brackets with "conducted in 2023/2024" or published in 2023/2024.

 - Line 73 - Specify that the study's aim is focusing on dispensing error rate related to GLP-1 RA.

 - Lines 74-76 - It is more appropriate for this information to be part of the discussion section.

 - Line 76 - References are not appropriately presented in the text, e.g., [4][7][24][25] should be [4, 7, 24, 25]. The same mistake occurs in several other places in the article.

 - Line 99, Line 101 -Replace "doctor" with "physician"

 - Replace "script" with "prescription".

 - Replace "script" with "prescription" in all places in the text of the article and the figure.

 - Line 116 - "Jira" - To ensure clarity for all readers, I recommend either explaining what Jira is or explicitly referring to it as "Jira software," as some readers may not be familiar with the term.

 - The flowchart requires revision to enhance its visual appeal. The current design employs excessive colours, and the overall quality is low.

 - Line 126 - Use "Physician" instead of "doctor"

 - Line 126 - Use "semaglutide" not "Ozempic"

 - Semaglutide should not be written in capital letters.

 - Conclusions should be written in separate section.

Author Response

Reviewer 1:

Comment - In both the study title and the study design, the term "case study" should be replaced with "cohort study."

Response: Thank you for this comment. We now refer to the study as a “cohort study” in both the title and study design section.

Comment - - It would be interesting for readers to provide data on obesity prevalence in Australia;

Response: Thank you for this important suggestion. We have now added the following sentence to the first paragraph of the introduction:

“Overweight and obesity rates in Australia have been rising steadily since 1995 and the latest census data revealed that nearly two-thirds (65.8%) of the adult population live with excess weight [1].”

Comment - - Line 36 - I recommend including more information regarding services provided by Ro and Eucalyptus to present their main mission in а summary.

Response: Thank you for this valuable recommendation. We have now added several sentences to paragraph 2 of the introduction that summarize Ro and Eucalyptus’ care models, their utility and their patient-facing mission statements. These sentences read as follows:

“Several of these services are comprehensive, coordinating lifestyle and pharmacological therapy through MDTs, such as Ro in the USA and Eucalyptus in Australia, Germany, Japan and the UK [7, 8]. However, most countries with DWLSs have a broad care model spectrum, with those at the minimalist end offering little more than access to prescriptions for weight-loss medications, including Glucagon-like peptide-1 receptor agonists (GLP-1 RA) [9]. While both Ro and Eucalyptus distinguish themselves from such care models by utilizing GLP-1 RAs as supplement to MDT-guided lifestyle therapy [10,8], there is evidence that some patients simply use the services to facilitate access to these medications [5]. Moreover, both service providers arguably centre more marketing content around the medications rather than care continuity. For example, the Eucalyptus homepage for men’s weight loss presents the following statement in bold text: “What medical weight loss offers is a fair go at better health” [11].  The Ro weight loss homepage is arguably a little more subtle, stating, “When it comes to weight loss, biology is your nemesis. Not willpower” [12], whereby the reader logically infers from surrounding content (and perhaps recent media exposure) that GLP-1 RAs address the genetic component of weight management. In both cases, the medications are presented within a social equity frame.”

Comment - - Line 59 and Line 70 - The Um et al. (2024)/The Um et al. (2023) - replace the year in brackets with "conducted in 2023/2024" or published in 2023/2024.

Response: Thank you for noticing this. We have now replaced the parenthesized publication date with “Published in 2024” in the first reference to the study like so:

“The best available benchmark for general dispensing safety appears to come from a global meta-analysis published in 2024. This meta-analysis pooled data from 62 studies across community, hospital…”

And we have removed the parenthesized publication date from the second reference to the study.

Comment - - Line 73 - Specify that the study's aim is focusing on dispensing error rate related to GLP-1 RA.

Response: Thank you for requesting this important detail. We have applied this change and the sentence now reads as follows:

“This study aimed to analyze the GLP-1 RA dispensing error rate of Australia’s largest comprehensive DWLS provider, Eucalyptus.”

Comment - - Lines 74-76 - It is more appropriate for this information to be part of the discussion section.

Response: Thank you for noticing this. We agree that this commentary should not be in the introduction and have now removed it. Our revised discussion now delivers this information in the ‘public health implications’ sub section through the following sentence:

GLP-1 RA-supported DWLSs have shown promise in making a significant contribution to a comprehensive solution [5,8,17,22,23].”

Comment - - Line 76 - References are not appropriately presented in the text, e.g., [4][7][24][25] should be [4, 7, 24, 25]. The same mistake occurs in several other places in the article.

Response: Thank you for noticing this. We have now updated the formatting of our references to align with the journal’s conventions, e.g. [4,7,24,25]

Comment - - Line 99, Line 101 -Replace "doctor" with "physician"

Response: Thank you for this comment. We have replaced the word “doctor” with “physician” on both of these lines.

 Comment - - Replace "script" with "prescription".

Response: Thank you for this comment. Each use of the word “script” throughout the manuscript has been replaced with “prescription”.

Comment - - Replace "script" with "prescription" in all places in the text of the article and the figure.

Response: Thank you for this comment. Each use of the word “script” has been replaced with “prescription” throughout the manuscript and figure.

Comment - - Line 116 - "Jira" - To ensure clarity for all readers, I recommend either explaining what Jira is or explicitly referring to it as "Jira software," as some readers may not be familiar with the term.

Response: Thank you for requesting this important detail. We have now refer to Jira as “Jira software”.

Comment  - The flowchart requires revision to enhance its visual appeal. The current design employs excessive colours, and the overall quality is low.

Response: Thank you for this vital request. We have now inserted a new flowchart, which was created through a digital mapping tool, Miro. The tool allowed us to create a more symmetrical and aesthetically pleasing flowchart.

Comment - Line 126 - Use "Physician" instead of "doctor"

Response: Thank you for this comment. The change has been applied.

Comment - Line 126 - Use "semaglutide" not "Ozempic"

Response: Thank you for this comment. The change has been applied.

Comment - Semaglutide should not be written in capital letters.

Response: Thank you for this comment. Each use of the word “semaglutide” is now non-capitalized.

Comment - Conclusions should be written in separate section.

Response: Thank you for noticing this. We have now inserted a “Conclusions” section at the end of the manuscript.  The section reads as follows:

“The findings from this study lay a vital foundation for ongoing research on dispensing safety in DWLSs and digital care in general. Despite the soaring uptake of GLP-1 RA-supported DWLSs in recent years, literature on the safety of such services remains scarce. GLP-1 RA dispensing errors represent one of the key dangers of such services, given both the relative potency of the medications and the outsourcing of their dispensation. This study’s findings suggest that comprehensive DWLSs have the potential to safely outsource GLP-1 RA dispensing to partner pharmacies via dedicated SaaS platforms. They do not, however, refute the argument that DWLSs enable large numbers of unsuitable patients to obtain Glucagon-like peptide-1 receptor agonists. To achieve such an end, further research is needed on dispensing and prescribing error rates of multiple comprehensive DWLSs.”

Reviewer 2 Report

Comments and Suggestions for Authors

SUGGESTIONS:

Introduction:

1. Authors should provide the clear research question and novelty of the study.

2. There is an over-reliance on secondary sources and broad generalizations about obesity and DWLSs without adequately connecting them to the specific context of GLP-1 RA dispensing errors.

3. The statement regarding the range of dispensing error rates (0 to 33%) from the meta-analysis by Um et al. (2024) lacks context. It is suggested for authors to explain the potential reasons for this wide range and its implications for the current study.

Materials and Methods:

4. The description of the retrospective case study design is unclear.  Authors should provide the inclusion and exclusion criteria. Also add more detail on the selection criteria for patients.

5. Have the participants completed an informed consent? It remains to include the study population, the sample and the sampling.

6. Please add software specification used in statistical analysis (ANOVA, Graph prism).

Results:

7. Results section can be improved in terms of write-up and representation of findings of the study.

Discussion:

8. Discussion section should be improved and according to the reported results. Authors must include a paragraph describing the public health implications

9. Limitation of the study should be discussed in limitation section of the manuscript.

10. Authors should add the Future perspective of the study. Manuscripts lack the implications of findings of the study. It would be more appropriate if authors add its clinical significance.

General Comments:

1. Authors should back their claims with recent references. The references are outdated and not comprehensive.

2. There are typographical errors throughout the manuscript e.g., "non-interference with patient experiences" should be "non-interference with patient experience". A thorough proofreading is necessary.

Comments on the Quality of English Language

There are typographical errors throughout the manuscript. A thorough proofreading is necessary.

Author Response

Reviewer 2:

Introduction:

Comment: Authors should provide the clear research question and novelty of the study.

Response: Thank you for requesting this important level of clarity. We have now inserted two new sentences and revised the final sentence of the last paragraph of the introduction to clarify these aspects of the manuscript. We now explain that the study lays an important foundation/benchmark in DWLS research and digital chronic care research in general, why this is important (huge uptake across a broad quality spectrum); and we reinforce the question: “the degree to which real-world comprehensive DWLSs can safely collaborate with third-party pharmacies to dispense GLP-1 RAs. The revised text now reads as follows:

“Given that the literature is yet to generate a benchmark for dispensing errors in digital chronic care, let alone GLP-1 RA dispensing errors in DWLSs, this study will lay a much-needed foundation for both fields. The importance of GLP-1 RA dispensing safety in DWLSs cannot be overstated in the current context in which increasingly large numbers of people are accessing the medications through a broad spectrum of online providers. The study’s findings should thus deliver an intuitive preliminary insight into the degree to which real-world comprehensive DWLSs can safely collaborate with third-party pharmacies to dispense GLP-1 RAs.”

Comment: There is an over-reliance on secondary sources and broad generalizations about obesity and DWLSs without adequately connecting them to the specific context of GLP-1 RA dispensing errors.

Response: Thank you for this insightful comment. We have now added 30 lines of text across paragraphs 2 and 3 of the introduction that add considerable detail to the context of GLP-1 RA dispensing errors in DWLSs. Firstly, we explain that while comprehensive DWLSs use GLP-1 RAs as a supplement to continuous MDT-led lifestyle therapy, their marketing tends to focus on the medications and there is evidence that some patients indeed use these services to simply access the medications. We then go onto explain that while there is consensus around the effectiveness of GLP-1 RAs, their potential (given the historical limitations of lifestyle interventions), and safety in clinical trial settings, there is no evidence to refute the concern of many stakeholders that DWLSs are enabling unsuitable patients to access the medications. We then clarify that GLP-1 RA dispensing represents one the two ways in which such safety errors can occur in DWLSs. Paragraphs 2 and 3 now read as follows:

“As a result of these access issues, multiple digital weight-loss services (DWLS) have started to emerge. Several of these services are comprehensive, coordinating lifestyle and pharmacological therapy through MDTs, such as Ro in the USA and Eucalyptus in Australia, Germany, Japan and the UK [7, 8]. However, most countries with DWLSs have a broad care model spectrum, with those at the minimalist end offering little more than access to prescriptions for weight-loss medications, including Glucagon-like peptide-1 receptor agonists (GLP-1 RA) [9]. While both Ro and Eucalyptus distinguish themselves from such care models by utilizing GLP-1 RAs as supplement to MDT-guided lifestyle therapy [10,8], there is evidence that some patients simply use the services to facilitate access to these medications [5]. Moreover, both service providers arguably centre more marketing content around the medications rather than care continuity. For example, the Eucalyptus homepage for men’s weight loss presents the following statement in bold text: “What medical weight loss offers is a fair go at better health” [11].  The Ro weight loss homepage is arguably a little more subtle, stating, “When it comes to weight loss, biology is your nemesis. Not willpower” [12], whereby the reader logically infers from surrounding content (and perhaps recent media exposure) that GLP-1 RAs address the genetic component of weight management. In both cases, the medications are presented within a social equity frame.  

Consensus around obesity’s genetic component and the role of GLP-1 RAs in modifying neural pathways involved in appetite is relatively well established [13, 14]. Additionally, clinical trials have consistently found GLP-1 RAs to be safe and highly effective in weight-loss cohorts [15, 16]. From an effectiveness perspective, a recent study showed that a GLP-1-RA-supported DWLS generated a mean 5-month weight loss (9.52%) than was over 3 times higher than that from a standalone lifestyle intervention (3.1%) [17]. Despite these discoveries and the knowledge that standalone lifestyle interventions have a low success rate [18,19], medical institutions continue to raise questions about the way DWLSs provide the medications. Namely, they appear concerned that certain DWLSs have inadequate safety protocols and allow people unsuitable for GLP-1 RAs to obtain such medications [20,21]. This an appropriate concern given that the effect of GLP-1 RA weight-loss therapy on unsuitable patients (e.g., patients who are not overweight) is unknown. Moreover, although GLP-1 RA-supported DWLSs have demonstrated good weight-loss and adherence outcomes across various cultures [8,9,17,22,23], they are yet to present any meaningful safety-related data. Stakeholders who question DWLS safety are ultimately concerned about errors that stem from one of two phases in the commencement of a patient’s care journey: the prescribing phase or the dispensing phase. “

Comment: The statement regarding the range of dispensing error rates (0 to 33%) from the meta-analysis by Um et al. (2024) lacks context. It is suggested for authors to explain the potential reasons for this wide range and its implications for the current study.

Response: Thank you for this excellent suggestion. We have now added five lines to the penultimate introduction paragraph to explain the reason for this variance. This revised text reads as follows:

“However, the study reported a dispensing error range from 0 to 33%, which is consistent with earlier meta-analyses and likely reflects considerable variance in pharmacy setting, technology use and denominator type [20]. Of these 3 factors, denominator type had the greatest impact on the wide dispensing error range across the 62 studies. Whereas some studies (n=5, 8.1%) used the total number of treated patients as the error rate denominator, others used the total number of doses (n=7, 11.3%) or dispensed items/orders (n=45, 72.6%).”

We have also added text to the endpoints section that justifies our choice of denominator type on the basis of the metanalysis findings.

The primary endpoint was the dispensing error rate, which was calculated by dividing the total number of dispensing errors by the total number of semaglutide orders. This denominator was selected on the basis of its prevalence in the available dispensing error literature [19].”

As a result of this revision, we have now changed the denominator in the analyses, which increased the observed error rates, from 0.076% to 0.35% in the total cohort, 0.089% to 0.41 in the female cohort, and 0.025 to 0.12 in the male cohort. These changes have been applied throughout the manuscript (including the abstract).

Materials and Methods:

Comment: The description of the retrospective case study design is unclear.  Authors should provide the inclusion and exclusion criteria. Also add more detail on the selection criteria for patients.

Response:

Thank you for this insightful comment. We state in the first sentence of the ‘participants’ section that the “study’s cohort comprised all Juniper and Pilot weight-loss patients who received at least one semaglutide order between 1 June and 1 December 2023”. However, we agree that we should include the specific inclusion and exclusion criteria physicians use to determine patient eligibility for the Semagltuide-supported program. We have now added these criteria to the participant section, which read as follows:

“The inclusion criteria were an initial body mass index (BMI) of 30 kg/m2 or greater (obesity) or a BMI of 27 kg/m2 or greater (overweight) in the presence of at least 1 weight-related comorbidity such as hypertension or sleep apnea. Key exclusion criteria included the following contraindications: acute pancreatitis; a personal or family history of medullary thyroid carcicnoma or in patients with multiple endocrine neoplasia type 2; a previous acute kidney injury; acute gallblader disease; hypoglycemia; a severe mental health condition; known hypersensitivity to Semaglutide or any of the product components; and patients with type 1 or type 2 diabetes. Eucalyptus physicians use their discretion in determining whether the semaglutide can be used concomitantly with other oral medications, which may interact with its gastric emptying effect.”

Comment: Have the participants completed an informed consent? It remains to include the study population, the sample and the sampling.

Response: Thank you for this comment. We have now copied and pasted the informed consent statement from the bottom of the manuscript at the end of the ‘participants’ section, which reads as follows: Eucalyptus patients consented to the service’s privacy policy at subscription, which includes permission to use their de-identified data for research.”

In regard to the study population, we have added a baseline characteristics table (Table 3) after the first paragraph of the results section and the following summary sentences to that paragraph:

“Across the sample of patients who received these orders, the mean age was 42.67 (±8.74) years and the mean BMI 33.47 (±5.82) kg/m2. Most patients were female (77.92%) and of Caucasian ethnicity (82.14%) (Table 3).”

Comment: Please add software specification used in statistical analysis (ANOVA, Graph prism).

Response: Thank you for noticing this omission. We have now added the following sentence to the end of the statistical analysis section:

“This test, and the analysis of all other study data were performed on R Studio (version 2023.06.1+524).”

Results:

Comment: Results section can be improved in terms of write-up and representation of findings of the study.

Response: Thank you for this important suggestion. We have now improved the content and organization of the results section. The first paragraph now focuses on the baseline characteristics of patients who received Semagltuide during the study period (and who were thus the cohort responsible for reporting dispensing errors). Three lines of text were added to this paragraph, along with a baseline characteristics table (Table 3). The paragraph now reads as follows:

“Between 1 June and 1 December 2023, the Eucalyptus DWLS dispensed 28165 semaglutide orders to Australian patients. Across the sample of patients who received these orders, the mean age was 42.67 (±8.74) years and the mean BMI 33.47 (±5.82) kg/m2. Most patients were female (77.92%) and of Caucasian ethnicity (82.14%) (Table 3).”

The second paragraph now focuses on dispensing error rates, both for the total cohort and individual programs. We have also added a Chi-square table (Table 4) to present the comparative results of the Pilot(male) and Juniper(female) programs. The second paragraph now reads as follows:

“A dispensing error was reported by patients in 99 of the 28165 orders over the study period, representing a total error rate of 0.35%. A subsequent analysis was conducted of the separate Eucalyptus Australia DWLSs to assess potential differences across male and female patients. Dispensing errors were reported in 92 (0.41%) of the 107477 semagltuide orders among the Juniper (female) cohort, and 7 (0.12%) of the 27654 orders in the Pilot (male) cohort. A chi-square test revealed that this slightly higher dispensing error rate observed in the female (Juniper) cohort was statistically significant, X2 (1, N = 135131) = 10.88, p < 0.001 (Table 4).”

The third paragraph of the results section is now dedicated to a breakdown of dispensing errors by their type and urgency rating. This paragraph is supported by table 5.

Discussion:

Comment: Discussion section should be improved and according to the reported results. Authors must include a paragraph describing the public health implications

Response: Thank you for this excellent recommendation. We have now divided the discussion into 4 sub-sections. The first contains 4 paragraphs and focusses on the significance of the study’s results. To clarify this significance, we added the following text (in bold) to the first paragraph:

This is the first study to report dispensing error rates in a GLP-1 RA-supported DWLS. Such services are becoming increasingly important throughout the world, as they arguably represent the most effective means of accessing continuous obesity care. Yet while previous research has demonstrated that DWLSs can increase access to continuous obesity care [5], the literature had hitherto failed to generate any evidence to silence the scepticism that they allow unsuitable patients to access such care [20,21]. GLP-1 RA medications have shown early promise in shifting the alarming trajectory of global obesity rates, but their effect on unsuitable weight-loss patients is unknown. Many stakeholders have raised appropriate concerns that DWLSs allow unsuitable patients, such as patients who are not overweight, to access GLP-1 RAs [20,21]. Their concern is thus that DWLSs are committing either GLP-1 RA prescribing or dispensing errors. This study investigated the GLP-1 RA dispensing error rate of Australia’s largest DWLS, establishing a foundation for ongoing DWLS safety research.”

The second sub section heeded your advice in explaining the public health implications of this research. This new sub section reads as follows:

“4.2 Public Health Implications

The above-discussed findings could have various public health implications. First and foremost, the observed dispensing error rates may be interpreted as preliminary evidence of the potential of GLP-1 RA-supported DWLSs to provide safe care. This will hopefully stimulate further research into other DWLS safety markers and efforts to develop national clinical governance standards for digital chronic care providers. Secondly, the study’s secondary endpoints may encourage the development of tighter monitoring mechanisms for digital care models that outsource medication dispensing. It was argued above that this responsibility should fall on DWLS providers, but it is feasible that this study and subsequent DWLS safety research will contribute to collaborative initiatives that promote transparency, such as public registries for GLP-1 RA prescribing and dispensing. Finally, this study adds another layer of nuance to the emerging literature on GLP-1 RA-supported DWLSs, which could feasibly lead to greater stakeholder confidence, government subsidization, and in turn, better public health outcomes. It is clear that traditional weight-loss services are incapable of addressing the global obesity problem [17,18,19]. GLP-1 RA-supported DWLSs have shown promise in making a significant contribution to a comprehensive solution [5,8,17,22,23]. However, public health systems need clear evidence of the safety of such services before they fully embrace them. This study represents the preliminary step towards a body of evidence that will allow an informed public safety assessment.”

The third sub-section provides suggestions for future research and reads as follows:

4.3 Future Research

Future research should aim to measure dispensing error rates in other GLP-1 RA-supported DWLSs to enable the eventual creation of an industry benchmark. Researchers should also start to analyze GLP-1 RA prescribing error rates in both DWLSs and F2F weight-loss programs. Case studies on clinical responses to medication errors in GLP-1 RA-supported DWLSs would add another important layer to the emerging field of literature.”

Comment: Limitation of the study should be discussed in limitation section of the manuscript.

Response: Thank you for noticing this omission. We have now added a “Strengths and Limitations” sub-header to the discussion (sub-section 4.4).

Comment: Authors should add the Future perspective of the study. Manuscripts lack the implications of findings of the study. It would be more appropriate if authors add its clinical significance.

Response: Thank you for this important suggestion. We have now added a dedicated sub-section to future research (sub-section 4.3), which reads as follows:

4.3 Future Research

Future research should aim to measure dispensing error rates in other GLP-1 RA-supported DWLSs to enable the eventual creation of an industry benchmark. Researchers should also start to analyze GLP-1 RA prescribing error rates in both DWLSs and F2F weight-loss programs. Case studies on clinical responses to medication errors in GLP-1 RA-supported DWLSs would add another important layer to the emerging field of literature.”

General Comments:

Comment: Authors should back their claims with recent references. The references are outdated and not comprehensive.

Response: Thank you for this comment. We have included 11 additional references to studies related to obesity in Australia [1] , GLP-1 RA-supported DWLSs [10,11,12,17,20,21,22,23], GLP-1 RA-induced weight loss [13,14] and standalone lifestyle interventions [18,19]. All 11 studies were published in the last 3 years, apart from the study explaining the science of GLP-1 RAs [13], which was published in 2016. The Um et al (2024) meta-analysis is the only recent study relevant to dispensing errors. However, as we explain in the introduction, even this study is of limited comparability to our research given that none of the pooled analyses were taken from Australian or modern digital care settings.   

Comment: There are typographical errors throughout the manuscript e.g., "non-interference with patient experiences" should be "non-interference with patient experience". A thorough proofreading is necessary.

Response: Thank you for noticing this. We applied the suggested change and several other typographical corrections such as to decapitalization of semaglutide. We subsequently ran the manuscript through Grammarly, which revealed that there were no further typographical errors.

Round 2

Reviewer 1 Report

Comments and Suggestions for Authors

I am satisfied with the author's responses to the concerns I raised in my initial review. However, I have a few minor comments regarding the figure:

  1. I recommend including the name of the software used to create the figure in its title.

  2. The term 'Doctor' should be replaced with 'Physician.'

  3. 'Semaglutide' is incorrectly capitalized and should be corrected."

Author Response

I am satisfied with the author's responses to the concerns I raised in my initial review. However, I have a few minor comments regarding the figure:

  1. I recommend including the name of the software used to create the figure in its title.

  2. The term 'Doctor' should be replaced with 'Physician.'

  3. 'Semaglutide' is incorrectly capitalized and should be corrected."

    Response:Thank you very much for noticing these shortcomings in Figure 1. We have now updated the flowchart with the suggested changes. "Doctor" has been replaced with "Physician"; "Semaglutide" has been uncapitalized; and we added the words "created with Miro software" to the figure title.

Reviewer 2 Report

Comments and Suggestions for Authors

Authors have diligently address all the comments.  

Author Response

Comment: Authors have diligently address all the comments.  

Response: Thank you very much for your clear feedback to the original submission, which greatly helped us improve our manuscript.